# Selective Photocatalytic Transformation of Lignin to Aromatic Chemicals by Crystalline Carbon Nitride in Water–Acetonitrile Solutions

**DOI:** 10.3390/ijerph192315707

**Published:** 2022-11-25

**Authors:** Meirou Huang, Huiqin Guo, Zhenxing Zeng, Hong Xiao, Hong Hu, Liu He, Kexin Li, Xiaoming Liu, Liushui Yan

**Affiliations:** 1Key Laboratory of Jiangxi Province for Persistent Pollutants Control and Resources Recycle, Nanchang Hangkong University, Nanchang 330063, China; 2Jiangxi Provincial Experimental Teaching Demonstration Center of Environmental Science and Engineering, Nanchang Hangkong University, Nanchang 330063, China; 3College of Environmental Science, Sichuan Agricultural University, Chengdu 611130, China

**Keywords:** crystalline carbon nitride, photocatalysis, lignin, aqueous solution effect, aromatic compounds

## Abstract

The photocatalytic conversion of lignin to aromatic compounds in aqueous solutions is a promising approach. We herein report a crystalline carbon nitride prepared by high-temperature thermal polymerization and alkali-metal molten salt treatment, which was then applied in the selective conversion of lignin to aromatic compounds. The results showed that the tri-s-tri-C_3_N_4_ presented a relatively high activity and selectivity for the conversion of lignin in aqueous solutions. In a 95% water–acetonitrile solution, it achieved a relatively high conversation rate of lignin, reaching 62.00%, and the selectivity of the C-C bond cleavage was high, at 86.8%. The characterization results obtained by TEM, UV-vis/DRS, PL, and transient photocurrent response showed that the ultra-high activity of tri-s-tri-C_3_N_4_ was mainly due to the improvements in crystallinity and light absorption. Mechanism studies showed that the dispersion of the catalyst and the combination of lignin and catalyst in aqueous solvents with different acetonitrile ratios were the main factors affecting lignin conversion. As the water content in the solutions increased, the primary active sites were converted from h^+^ to ·O_2_^−^. This study revealed the interactions between lignin, photocatalysts, and reaction solutions, providing a theoretical basis for the photocatalytic conversion of lignin in aqueous solutions.

## 1. Introduction

The large-scale development and efficient utilization of biomass are an effective way to solve the global energy crisis and the environmental problems caused by fossil fuels, but they are still challenging for scientific and industrial communities [1]. Lignocellulose is an important component of natural biomass energy. In particular, lignin, a naturally occurring sustainable aromatic polymer, can provide high-value-added aromatic chemicals or fuels as an alternative to fossil resources [2]. The three-dimensional network structure of lignin consists of phenylpropane units connected by C-O and C-C bonds [3,4]. Usually, the cleavage of C-O bonds can only produce aromatic monomers with a theoretical yield of no more than 50% [5]. Thus, it is essential to develop efficient methods for the cleavage of the C-C bonds for higher yields of aromatic chemicals. However, the high bond dissociation energies of the C-C bond mean that this process must be conducted under harsh conditions, and low quantities of aromatic monomer molecules are obtained [6]. The key to the preparation of high-value-added chemicals from lignin is the efficient selective cleavage of C-C bonds under mild conditions.

Semiconductor photocatalysis has recently been reported as a promising approach to the selective cleavage of C-C bonds from lignin [7,8]. The conversion of lignin into high-value-added aromatic chemicals has been studied using semiconductors as photocatalysts. For example, Zhang’s group [9] reported the transformation of lignin in acetonitrile solvent using commercially available CeCl_3_ as a photocatalyst. Highly selective C-C bond cleavage accompanied by a high yield of aldehydes and N-containing products was achieved. Wang and coworkers [10] disclosed the mechanism behind the cleavage of C-C bonds using a VO(acac)_2_ photocatalyst. However, these works used metal catalysts, which easily agglomerate and dissolve in water and lead to a decrease in the active area and a loss of components [11,12]. The development of metal-free catalysts with high visible-light absorption, a low cost, and abundant catalytic active sites is necessary for lignin transformation [13].

Wang’s group [14] prepared and successfully applied visible-light-responsive mesoporous graphitic carbon nitride (mpg-C_3_N_4_) for lignin conversion in acetonitrile solvent. The results showed that this method achieved a high selectivity of 91% for C-C bond cleavage. Recently, our group prepared S-doped mesoporous carbon nitride (MSCN-0.5) [15] and cyano-sulfur co-modified carbon nitride (MCSCN) [16] by direct calcination and applied these substances for the conversion of different lignins. A high conversion rate and highly selective C-C bond cleavage in acetonitrile solvents were achieved. Considering the disadvantages of carbon nitride synthesized by direct high-temperature thermal polycondensation, i.e., the low specific surface area and visible-light responsiveness [17,18], the crystalline structure of carbon nitride, which presents fewer charge recombination sites and a stronger photocatalytic performance, has attracted much attention [19]. Carbon nitride has been extensively applied in the degradation of pollutants and many organic synthesis reactions [20,21]. Therefore, its application in the photocatalytic conversion of lignin is expected to be effective.

In addition to catalysts, another issue that deserves attention is the fact that most studies on the photocatalytic conversion of lignin have been conducted using organic solvents. From the perspective of green chemistry, it is necessary to explore the conversion behavior and mechanisms of lignin in aqueous solutions. Recently, Shao et al. [22] reported the C_β_-O bond cleavage of lignin in an aqueous solution and found that the selectivity of the aromatic monomers and the lignin conversion rate increased by 170% and 58%, respectively, compared with under anhydrous reaction conditions. Their mechanistic studies showed that water functioned as a hydrogen donor and promoted the hydrogenation of the C_β_-O bonds in lignin by overcoming the limitation of the proton supply. This work provided a new perspective on the green transformation of lignin. However, few reports have been published in this area, and the selective cleavage of C-C and C-O bonds in aqueous solutions is of great significance for the conversion and utilization of lignin.

In the current research, carbon nitride with a high crystallinity was prepared by the alkali-metal molten salt strategy. The properties of the crystal-state structure, including the morphology, physicochemical properties, photoelectric properties, and band structure, were systematically studied. Based on the photocatalytic transformation of lignin in water–acetonitrile solutions, we attempted to construct an aqueous solution system for lignin conversion. The β-O-4 lignin model compound was taken as the substrate, and the conversion mechanism was studied. The results are expected to provide a theoretical basis for the conversion of lignin in aqueous solutions.

## 2. Materials and Methods

### 2.1. Preparation of Materials

For the preparation of g-C_3_N_4_, 10 g of melamine (Aladdin, 99% pure) was deposited into a 50 mL ceramic crucible with a cover and placed into a muffle furnace. The furnace was heated at a rate of 5 °C/min to 550 ℃, and this temperature was held for 4 h. After natural cooling to room temperature, a yellow powder was obtained, i.e., g-C_3_N_4_.

For the preparation of tri-C_3_N_4_, 1.2 g melamine was fine-ground with a eutectic mixture (5.4 g LiCl (Aladdin, ≥99.0% pure) and 6.6 g KCl (Aladdin, 99.5% pure)) then transferred to a 50 mL alumina crucible and placed in a vacuum tube furnace. The mixture was heated to 550 °C at a heating rate of 5 °C/min under an Ar atmosphere and calcined at this temperature for 4 h. After the sample cooled to room temperature, it was repeatedly washed with boiling water and dried overnight in a vacuum-drying oven at 60 °C.

For the preparation of tri-s-tri-C_3_N_4_, 10 g of melamine was placed into a 50 mL alumina crucible with a cover, heated to 500 °C at a heating rate of 5 °C/min in a muffle furnace, and held at this temperature for 4 h. Subsequently, a mixture of 1.2 g pretreated melamine (melon), 5.4 g LiCl, and 6.6 g KCl was ground in a mortar, transferred to a 50 mL alumina crucible, and placed in a vacuum tube furnace, where it was heated to 550 °C at a rate of 5 °C/min under an Ar atmosphere and calcined for 4 h at this temperature. After cooling to room temperature, the mixture was washed repeatedly with boiling water and dried at 60 °C in a vacuum.

### 2.2. Material Characterization

The morphology and structure of the samples were characterized by SEM (ZEISS Sigma 300) and TEM (JEOL JEM 2100F, accelerating voltage of 200 kV). The crystal structures of the samples were analyzed by XRD (D8 ADVANCE) with Cu-K_α_ at 40 mA and 40 kV. The chemical structures of the samples were investigated using FT-IR (Bruker VERTEX 70 FT-IR). The valence band spectra of the samples were determined by XPS (Thermo Scientific K-Alpha). In addition, the UV-vis diffuse reflectance spectra of the materials were measured on a UV-vis/DRS (Lambda 750S UV-VIS/NIR), with a wavelength range of 200~800 nm and the bandgaps (*E_g_*) of the materials converted using the Kubelka–Munk formula. To explore the influence of the crystalline structure of the samples on their optical properties, the separation efficiency of the photogenerated carriers on the surface of the materials was investigated using a fluorescence spectrophotometer (Hitachi F-7000, excitation wavelength 330 nm) and a transient photoluminescence spectrometer (Edinburgh FS5). The specific surface areas and pore size distributions of the materials were determined using a specific surface area and porosity analyzer (Quantachrome NOVA 2000e). Transient photocurrents and Mott–Schottky curves were measured on a CHI660E electrochemical workstation (Chenhua, China) using the traditional three-electrode system.

### 2.3. Photocatalytic Experiments

The photocatalytic conversion reactions were all conducted in a double-layer photocatalytic glass reactor, and a 425 nm LED lamp was used as the light source under an air atmosphere and room temperature conditions. To exclude the influence of temperature on the reaction system, the water flowed continuously in the reactor during the reaction to maintain a temperature equilibrium within the reactor. The operation steps of the photocatalytic conversion of the lignin model compound (2-phenoxy-1-phenylethanol (Aladdin, 98% pure)) were as follows: First, we transferred 50 mL 0.4 mmol/L of the lignin model compound solution to the photocatalytic reactor and added 10 mg photocatalyst after ultrasonic dispersion for 5 min to mix thoroughly. The mixtures were magnetically stirred in the dark for 30 min before the light source was turned on in order to reach adsorption equilibrium. The reactants were continuously stirred by magnetic forces throughout the whole photocatalytic reaction. Samples were collected (1.0 mL for each sample) at given intervals using a 1 mL pipette gun. Then, they were centrifuged to separate them from the catalyst and filtered through a 0.22 μm PECT filter head. After that, the samples were analyzed and detected by GC-MS (Agilent 7890A/5975C, USA) and HPLC (Agilent 1100 Series, Santa Clara, CA, USA).

## 3. Results and Discussion

### 3.1. Morphology Analysis

Melamine was first condensed at a temperature of 500 °C with a heating rate of 5 °C/min to obtain treated melamine (melon). The melon was then polymerized into heptazine-phase carbon nitride (tri-s-tri-C_3_N_4_) with a high crystallinity under an alkali-metal molten salt environment; melamine could be used to synthesize triazine-phase carbon nitride (tri-C_3_N_4_) in the same way, as shown in Appendix A. The micromorphology of the prepared g-C_3_N_4_, tri-C_3_N_4_, and tri-s-tri-C_3_N_4_ was characterized by SEM. As shown in Figure 1, we observed that the agglomeration of the g-C_3_N_4_ was due to direct high-temperature heat-shrinkage polymerization. The tri-C_3_N_4_ and tri-s-tri-C_3_N_4_ synthesized from melamine and melon treated with molten salt showed nanodot and nanostick morphologies, respectively. This was due to the low degree of melon polymerization, meaning that the surface was rich in unpolymerized amino groups. These unpolymerized amino groups grew within the voids of the molten salt and eventually formed a nano-short-rod morphology. This irregular nano-short-rod structure provided a longer optical path for light harvesting and more active sites, thus benefiting the photocatalytic performance of the material.

### 3.2. Characterization of Physicochemical Properties

To confirm the successful synthesis of crystalline carbon nitride, the constructed materials were further characterized by XRD, FT-IR, HRTEM, and BET. XRD was used to analyze the crystal structure of the three materials, and the results are shown in Figure 2a. Two obvious diffraction peaks at around 2θ = 13.0° and 2θ = 27.4° were identified as typical diffraction peaks of g-C_3_N_4_, corresponding to the (100) crystal plane of the in-plane repeated heptazine structural unit and the (002) crystal plane of the layered, stacked, and conjugated aromatic structure [23]. Improving the crystal state would make the sheet layer of the material more stretched. This was the main reason for the small angle shift of the diffraction peak corresponding to the (100) crystal plane of tri-C_3_N_4_ and tri-s-tri-C_3_N_4_. In addition, the stretching of the sheet layer was conducive to the stacking of the sheets, resulting in a decrease in the layer spacing, which was consistent with the right shift of the diffraction peak corresponding to the (002) crystal plane of tri-s-tri-C_3_N_4_ [24]. The XRD spectrum of tri-C_3_N_4_ contained many sharp diffraction peaks between 10 and 40°, which were typical features of tri-C_3_N_4_′s high crystallinity. However, the high-temperature alkali-metal molten salt treatment may have had a certain peeling effect on the materials, resulting in the weakening of the XRD diffraction peak strength of tri-C_3_N_4_ and tri-s-tri-C_3_N_4_. To explore the effect of the alkali-metal molten salt treatment on the material skeleton, FT-IR was used to further investigate the chemical structures of the samples, and the results are shown in Figure 2b. All materials showed the typical carbon nitride skeleton absorption peaks, indicating that the alkali-metal molten salt treatment did not destroy the basic structure of the materials. Compared with g-C_3_N_4_, the absorption peak strength of the material treated with alkali-metal molten salt at 3200 cm^−1^ was significantly reduced, implying a decrease in the number of amino groups and indicating that the alkali-metal molten salt treatment improved the degree of carbon nitride polymerization [16].

The microscopic morphology of g-C_3_N_4_, tri-C_3_N_4_, and tri-s-tri-C_3_N_4_ was observed by HRTEM. As shown in Figure 3a–c, g-C_3_N_4_ showed a relatively smooth, layered stacking topography; tri-C_3_N_4_ presented a sheet-structured stacking morphology composed of nanodots; and tri-s-tri-C_3_N_4_ demonstrated a stacked nanorod structure. After adjusting the magnification of the TEM, we could observe that tri-C_3_N_4_ and tri-s-tri-C_3_N_4_ contained crystal lattices after the alkali-metal molten salt treatment, while there were no crystal lattices in g-C_3_N_4_. This further proved the successful fabrication of tri-C_3_N_4_ and tri-s-tri-C_3_N_4_ with a crystalline structure.

The lattice fringes were calibrated by Digital Micrograph software, producing results of 0.3458 nm and 1.0574 nm for tri-C_3_N_4_ and tri-s-tri-C_3_N_4_, respectively. The determined lattice fringes corresponded to the interlayer distance of the (002) plane. Meanwhile, the diffraction spots and diffraction fringes could be observed in the electron diffraction patterns of tri-C_3_N_4_ and tri-s-tri-C_3_N_4_, but not in that of g-C_3_N_4_. The high crystallinity of tri-C_3_N_4_ and tri-s-tri-C_3_N_4_ has been proven [25]. This higher crystallinity reduced the charge recombination center caused by the defect and increased the separation efficiency of the photogenerated electrons and holes. In addition, the expansion of the conjugated structure and the increased structural ordering reduced the obstacles to the charge movement process, allowing more photogenerated carriers to participate in the catalytic reaction on the surface of the material.

The surface area and pore structure of the catalysts had a great influence on the catalytic reaction. The surface area and pore-size distribution of the materials were characterized by nitrogen adsorption–desorption instruments. It can be seen from Figure 4a that the adsorption–desorption isothermal curves of all samples showed a typical type IV isotherm with an H1-type hysteresis loop, indicating that there were mesopores in all the materials. Combined with the TEM results, the mesoporous structure of g-C_3_N_4_ was derived from the gap between the carbon nitride layers. The mesoporous structure of tri-C_3_N_4_ and tri-s-tri-C_3_N_4_ was mainly derived from the improvement in the carbon nitride crystal state in addition to the interlayer gap. After the analysis of the adsorption–desorption curves, it could be seen that the surface areas of g-C_3_N_4_, tri-C_3_N_4_, and tri-s-tri-C_3_N_4_ were 46.250 m^2^g^−1^, 69.914 m^2^g^−1^, and 97.175 m^2^g^−1^, respectively. The surface areas of the crystalline materials were greatly improved compared with that of g-C_3_N_4_, indicating that the pore structure of the materials’ surfaces was significantly increased after the alkali-metal molten salt treatment. The BJH model was used to analyze the pore structure of the materials, as shown in Figure 4b. We found both a mesoporous and a macroporous structure in tri-s-tri-C_3_N_4_. The pore capacities of g-C_3_N_4_, tri-C_3_N_4_, and tri-s-tri-C_3_N_4_ were 0.1204 cm^−3^g^−1^, 0.1596 cm^−3^g^−1^, and 0.2620 cm^−3^g^−1^, respectively. Thus, tri-s-tri-C_3_N_4_ had the largest pore capacity, which coincided with its high specific surface area. The increase in the surface area and porosity facilitated the mass transfer process of the catalytic reactions.

### 3.3. Optoelectronic Properties

The spectral absorption properties of the materials were characterized by ultraviolet-visible/diffuse reflectance (UV-vis/DRS) spectroscopy. As shown in Figure 5a, the crystalline materials exhibited significant spectral absorption in the wavelength range of 200 nm–420 nm, corresponding to the transition of valence-band electrons (N 2p orbitals) of carbon nitride to C 2p orbitals [26]. Secondly, the light-absorption capacity of the crystalline materials in the ultraviolet and visible light ranges (250 nm–800 nm) was significantly higher than that of g-C_3_N_4_. This indicated that the improvement of the crystalline state increased the light-absorption capacity of the materials. This was because the increase in the degree of carbon nitride polymerization was conducive to the transition of π electrons from bonded orbitals to antibonding orbitals. The separation performance of the photogenerated carriers affected the efficiency of the electron transfer between the catalyst and the reaction substrate. Fluorescence spectrophotometers were used to explore the separation efficiency of the photogenerated carriers in all the materials. As shown in Figure 5b, at an excitation wavelength of 330 nm, g-C_3_N_4_ produced a strong fluorescence emission peak in the wavelength range of 350 nm−600 nm, indicating a high level of photogenerated electron–hole recombination in g-C_3_N_4_. The fluorescence emission peak intensity of the crystalline materials was lower than that of g-C_3_N_4_, with that of tri-s-tri-C_3_N_4_ being the lowest. This indicated that tri-s-tri-C_3_N_4_ had the highest photogenerated hole–electron separation efficiency and a greater ability to inhibit photogenerated carrier recombination. The improved crystallinity of the materials reduced the number of charge recombination sites (unpolymerized amino acids) on the surface, thereby inhibiting charge recombination.

In addition, the charge recombination processes of g-C_3_N_4_, tri-C_3_N_4_, and tri-s-tri- C_3_N_4_ were characterized by transient fluorescence spectroscopy; the results are shown in Figure 6a, and the fitted data are shown in Appendix A. The short-lifetime component (τ_1_) is usually determined by the nonradiative relaxation associated with material defects, while the long-lifetime component (τ_2_) can be attributed to the radiation generated by photogenerated carrier recombination [27]. Compared with g-C_3_N_4_ (29.6577 ns), the long-lifetime components (τ_2_) of tri-C_3_N_4_ (5.3558 ns) and tri-s-tri-C_3_N_4_ (3.2833 ns) were reduced, and that of tri-s-tri-C_3_N_4_ was the lowest. This indicated that the number of photogenerated hole–electron pairs in tri-s-tri-C_3_N_4_ was lower than that in g-C_3_N_4_ and tri-C_3_N_4_. The interfacial separation process of most charges in tri-s-tri-C_3_N_4_ shortened the fluorescence lifetime. To confirm this finding, the materials were characterized by transient photocurrents, and the results are shown in Figure 6b. The transient photocurrent response of tri-s-tri-C_3_N_4_ was enhanced compared with that of g-C_3_N_4_ and tri-C_3_N_4_, indicating that the construction of the heptazine crystal phase was more conducive to improving the charge separation efficiency. This result corresponded well with the results of the PL spectroscopy and TR-PL spectroscopy.

### 3.4. Photocatalytic Performance

*Camellia oleifera* shell lignin contains abundant β-O-4 structural fragments. Because of the complex aromatic structures and interlinkages of lignin, its depolymerization into high-value chemicals is still a challenge for researchers. Nowadays, lignin model compounds are commonly used as research objects to explore the cleaving performance and mechanisms of catalytic systems and catalysts for lignin linkages. As a representative β-O-4 lignin model compound, 2-phenoxy-1-phenylethanol **1** (referred to as ML) contains typical lignin linkages (Appendix A). Thus, it is widely used in research on the conversion of lignin.

In this work, ML was initially used as the object to explore the effects of different crystalline carbon nitride structures on substrate conversion, product yield, and the selective cleavage of C_α_-C_β_ bonds in acetonitrile solvent. The photocatalytic reaction products of ML were qualitatively analyzed by GC-MS, and the results are shown in Figure 7a. It was found that the photocatalytic products of ML were benzaldehyde (**2**) (Aladdin, ≥98% pure) and 2-phenoxyacetophenone (**4**) (Aladdin, >98% pure). The GC-MS instrument was not sensitive to certain thermally unstable substances, so HPLC was used to further analyze the reaction products, and the results are shown in Figure 7b. In addition to the two products detected by GC-MS, phenyl formate (**3**) (Aladdin, 95% pure); formic acid; benzoic acid; and p-benzoquinone were also detected. Benzaldehyde and phenyl formate were C_α_-C_β_ bond-cleavage products, and 2-phenoxyacetophenone was produced by the direct oxidation of the C_α_-OH bonds in ML. Formic acid (Aladdin, ≥98% pure); benzoic acid (Aladdin, 99.5% pure); and p-benzoquinone (Aladdin, ≥99.5% pure) may have been by-products of the excessive oxidation of the other products, with p-benzoquinone resulting from the excessive oxidation of phenol in air. Quantitative analysis was mainly carried out for products (**2**), (**3**), and (**4**).

The effects of light, photocatalyst, and solvent conditions on the photocatalytic conversion of ML were further explored, and the results are shown in Figure 8 and Figure 9 and Table 1. First, in groups 1 and 8, the ML was extremely stable and could not be converted without a catalyst or light. In comparison, in group 4, tri-s-tri-C_3_N_4_ had the best photocatalytic activity against ML. The conversion of ML reached 100% (about 6.53 times that of g-C_3_N_4_), corresponding to the characterization results of tri-s-tri-C_3_N_4_, which showed a high photogenerated carrier separation efficiency, high surface-charge migration efficiency, and large specific surface area. Although tri-C_3_N_4_ presented high crystallinity, the insufficient ductility of the aromatic ring led to a lower photogenerated charge yield, and its catalytic activity was comparable to that of g-C_3_N_4_. Although tri-s-tri-C_3_N_4_ had reduced selectivity for C_α_-C_β_ bond cleavage compared to g-C_3_N_4_ and tri-C_3_N_4_, its conversion to substrates was greatly improved, and it maintained high selectivity. In addition, by comparing the results of groups 4, 5, 6, and 7, it was found that the proton-donation capacity of the solvent had a huge impact on the reaction. The conversion rate of ML increased under the same conditions with the weakening of the proton-donation capacity of the organic solvents, which were ranked in terms of proton-delivery capacity as follows: methanol > acetone > ethyl acetate > acetonitrile. For the methanol, acetone, and ethyl acetate solvents with high proton-donation capacities, the conversion rate of ML was 8.62%, 36.76%, and 42.63%, respectively. In the acetonitrile solvent, which had the weakest proton-donation ability, tri-s-tri-C_3_N_4_ was most effective at photocatalytically converting ML (Appendix A). The selectivity of the C_α_-C_β_ cleavage also differed between organic solvents, though they all demonstrated high selectivity for the substrates. In addition, to test the recoverability of the catalysts, catalyst-cycle experiments were carried out. The results showed that tri-s-tri-C_3_N_4_ had a stable structure and excellent photocatalytic activity after four cycles, with the conversion rate of ML still reaching 94.65% (Appendix A).

To fabricate a greener and more economical reaction solvent, we further explored the photocatalytic conversion of ML in water–acetonitrile solutions with different volume ratios. The results are shown in Figure 10 and Table 2. We found that when a small volume of water was added, the conversion rate of ML decreased sharply. With an increase in the proportion of the aqueous phase, the conversion rate slowly increased to a stable level. However, compared with pure acetonitrile, the conversion rate in the mixed solvents was low, and the yield of benzaldehyde was consistent with the trend of the conversion rate. The conversion rate of ML, the yield of benzaldehyde, and the C_α_-C_β_ bond-cleavage selectivity were reduced under a high aqueous phase volume ratio, which not only substantially decreased the solvent cost but also lessened the environmental hazards.

### 3.5. The Photocatalytic Conversion Mechanism

The band structure of the photocatalytic materials was explored, and the results are presented in Figure 11. As shown in Figure 11a, the bandgap widths of g-C_3_N_4_, tri-C_3_N_4_, and tri-s-tri-C_3_N_4_ were 2.48 eV, 2.74 eV, and 2.55 eV, respectively. This indicated that these materials had a visible light response. The XPS valence band test results (Figure 11b) showed that the valence band maximum values of g-C_3_N_4_, tri-C_3_N_4_, and tri-s-tri-C_3_N_4_ were 2.46 eV, 2.11 eV, and 2.39 eV, respectively. Due to the inherent error of the XPS VB test, attributed to the contact potential difference between the samples and the analyzer, the valence band value only represents the relative position [28].

To accurately assess the band structure, the flat band potentials of g-C_3_N_4_, tri-C_3_N_4_, and tri-s-tri-C_3_N_4_ were further calibrated using a Mott–Schottky diagram with a frequency of 1000 Hz, and the results are shown in Figure 11c. The flat band potentials of the materials were −1.48, −1.50, and −1.42 eV, respectively, and the slope value of the Mott–Schottky curve was positive, indicating that all the catalysts were n-type semiconductors [29]. Considering the *E_g_* values of the materials, the conduction band minimum potential (*E_CBM_*) of these n-type semiconductors was 0.1 V lower than the *E_fb_* value [30]. Thus, the *E_VBM_* values of the materials were calculated, and the results are shown in Figure 11d. We found that the *E_VBM_* value (1.23 eV) of tri-s-tri-C_3_N_4_ was higher than that of H_2_O/OH, indicating that ·OH would not be produced in the photocatalytic reactions by tri-s-tri-C_3_N_4_. However, the *E_VBM_* value (1.23 eV) was larger than the oxidation potential required for H_2_O/O_2_ (0.82 eV), suggesting that the addition of water would deplete the photogenerated holes and produce O_2_. The *E_CBM_* value (−1.32 eV) of tri-s-tri-C_3_N_4_ was higher than that of the reduction potential (−0.33 eV) required by O_2_/O_2_^−^, suggesting that tri-s-tri-C_3_N_4_ could transfer electrons to O_2_ to produce ·O_2_^−^ [31].

The effects of the free-radical-quenching agents on the reactions were also explored, and the results are shown in Figure 12. We found that the photocatalytic reaction was not inhibited when using *tert*-butyl alcohol (t-BuOH) as the ·OH quenching agent, which suggested that ·OH was not the main active group in the reactions using pure acetonitrile and water–acetonitrile (30/70, 90/10, *v/v*) solutions. Notably, the conversion rate slightly improved under a higher proportion of water. This result was attributed to the transformation of the alcohol radicals generated by t-BuOH into hydroxyl radicals, which accelerated the oxidative depolymerization of the lignin [32].

When p-benzoquinone (p-BQ) was used as the O_2_^−^ quenching agent, both the conversion rate of the substrate and the product yield in the acetonitrile solvent were slightly reduced. This indicated the presence of ·O_2_^−^ in the process, which also suggested that tri-s-tri-C_3_N_4_ could transfer electrons to oxygen to produce ·O_2_^−^. However, the addition of benzoquinone to the water–acetonitrile (30/70, *v/v*) solvent accelerated the conversion of the lignin, but the yield of benzaldehyde was not improved accordingly. This was probably because the addition of water increased the solubility of oxygen in the reaction solutions, and some of this oxygen was reduced to ·O_2_^−^ by photogenerated electrons, resulting in increased levels of ·O_2_^−^. The consumption of electrons by oxygen would indirectly promote the separation of photogenerated carriers. All of the above would be beneficial to the conversion of lignin. In the water–acetonitrile (90/10, *v/v*) solution, the photocatalytic reaction was severely inhibited, which further indicated that ·O_2_^−^ was the primary active group in these reactions.

When triethanolamine (TEA) was used as the quenching agent for the holes (h^+^), the photocatalytic reaction was inhibited in both the acetonitrile and acetonitrile–water solutions. Additionally, a more significant inhibition effect was demonstrated in the solvent containing a high proportion of acetonitrile. This contributed to the fact that compared with the lignin molecules, TEA was more likely to react with the photogenerated holes, which hindered the electron transfer between the lignin and the holes, resulting in the inhibition of the photocatalytic reactions.

The abovementioned results suggested that in these reactions, photogenerated holes played a primary role when solvents with a high proportion of acetonitrile were used, and ·O_2_^−^ played an auxiliary role. As for the solvents containing a high proportion of water, ·O_2_^−^ played a leading role and h^+^ played an auxiliary role.

To verify the above hypothesis, a series of substrates were further used to explore the photocatalytic reaction path and deprotonation site of the lignin model compound, and the experimental results are shown in Appendix A. It can be seen that in group **1,** the conversion rate of benzaldehyde was high, and the reaction system promoted its oxidation. Therefore, it is necessary to control the reaction conditions to prevent the excessive oxidation of the product. The conversion rate of phenyl formate in group **2** was extremely low, indicating that phenyl formate was stable in the reaction system and not easily oxidized. This further proved the cleavage of the C_α_-C_β_ bonds of ML under experimental conditions. In group **3**, the conversion rate of 2-phenoxyacetophenone was very low, indicating that 2-phenoxyacetophenone was not a reaction intermediate and C_α_-H was not a deprotonation site for ML. In [14], Wang et al. used isotopic labeling substrates to perform reactions and found that the oxidation of C_β_-H played a major role in the substrate’s transformation, which also indicated that the deprotonation site of C_α_-C_β_ bond cleavage was located at C_β_-H. In group **4**, it can be seen that C_α_-OH played an essential role in the conversion. The above analysis was perfectly consistent with the assumptions of the previous experiments.

The conversion mechanism of ML in acetonitrile solvent was proposed according to the above findings and is shown in Figure 13. Taking ML as an example, the conversion process comprised the following four steps: Firstly, the carbon nitride was excited by blue light at 425 nm, generating holes and electrons in the valence and conduction bands, respectively. The C_β_-H bonds in ML were deprotonated under the action of the photogenerated holes, which produced the C_β_ radical intermediate **A**. Then, H^+^ and O_2_ in the air were reduced by electrons to generate H· and ·O_2_^−^, respectively. Transition-state intermediate **B** was generated by combining **A** with ·O_2_^−^. **B** combined with H· to form the transition-state intermediate **C**. Afterwards, **C** was dehydrated through intramolecular electron transfer to induce C_α_-C_β_ bond cleavage to form benzaldehyde and phenyl formate. Meanwhile, the C_α_-H bonds were also oxidized by the photogenerated holes to form **4**. This was consistent with the mechanism of the cleavage of C_α_-C_β_ in 2-phenoxy-1-phenylethanol by carbon nitride according to Wang et al. [14].

When water was added to the system, the amino groups on the catalyst surface formed hydrogen bonds with the water molecules. Compared with the substrate, the water molecules were more likely to capture photogenerated holes and be oxidized to generate O_2_ and H^+^, which reduced the activation of H at the C_β_ site in ML and the formation of the C_β_ radical intermediate **A**, thereby inhibiting the progress of the reaction. With an increase in the proportion of water added, the dispersibility of the carbon nitride in the solvent was improved, and the water molecules and substrate molecules competed for holes in the catalyst, decreasing the conversion rate. In contrast to the acetonitrile solvent, the water molecules in the mixed solvents consumed holes to inhibit the formation of the C_β_ radical intermediate A. This confirmed that the conversion of the substrate and the product yield were reduced in the presence of water molecules. Combined with the above analysis, the added water molecules competed with the substrate molecules to consume holes, and the generated O_2_ molecules and H^+^ were reduced by electrons to form ·O_2_^−^ and H·. The formation of ·O_2_^−^ and H· promoted the conversion of the transition intermediates **A** and **B** and the generation of products.

## 4. Conclusions

In conclusion, tri-s-tri-C_3_N_4_ was prepared by the secondary treatment of melon with alkali-metal molten salt. The high crystallinity of g-C_3_N_4_ substantially reduced the number of charge recombination sites and enhanced the charge mobility, improving its photocatalytic performance. A green photocatalytic strategy was developed for the C-C bond cleavage of lignin models in aqueous solutions using tri-s-tri-C_3_N_4_ as the catalyst. The β-O-4 linkages in the lignin were efficiently cleaved into aromatic aldehydes with a high yield and selectivity. The dispersibility and combination of lignin and catalyst in the solvents containing acetonitrile and water at different volume ratios were the main influencing factors, according to the trapping experiments and characterization results. This study provides a new strategy for the conversion of lignin in green solutions.

## Figures and Tables

**Figure 1 ijerph-19-15707-f001:**
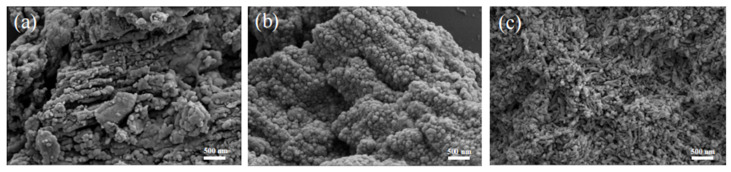
SEM images of g-C_3_N_4_ (**a**), tri-C_3_N_4_ (**b**), and tri-s-tri-C_3_N_4_ (**c**).

**Figure 2 ijerph-19-15707-f002:**
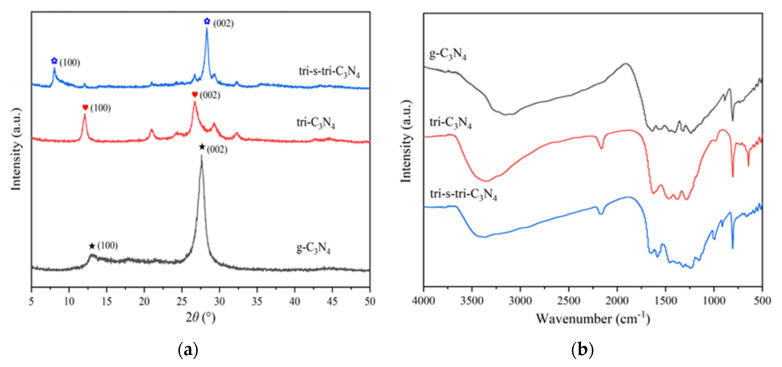
XRD pattern (**a**) and FT-IR spectra (**b**) of g-C_3_N_4_, tri-C_3_N_4_, and tri-s-tri-C_3_N_4_.

**Figure 3 ijerph-19-15707-f003:**
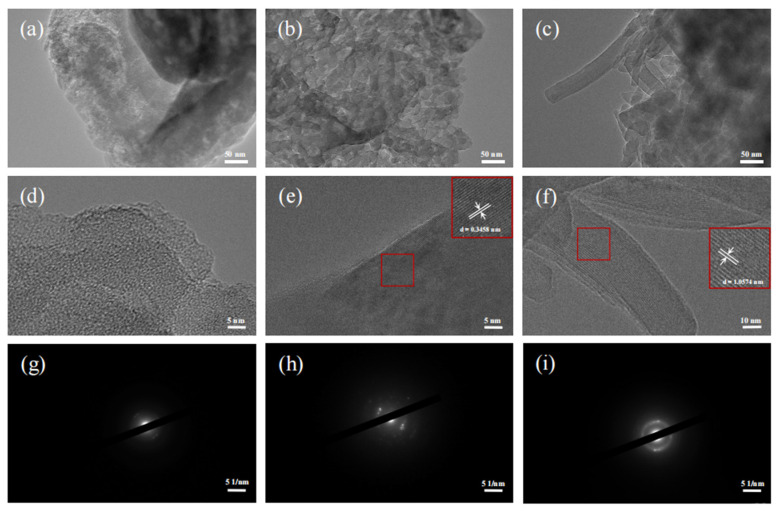
TEM images (**a**,**d**) and electron diffraction patterns (**g**) of g-C_3_N_4_; TEM images (**b**,**e**) and electron diffraction patterns (**h**) of tri-C_3_N_4_; TEM images (**c**,**f**) and electron diffraction patterns (**i**) of tri-s-tri-C_3_N_4_.

**Figure 4 ijerph-19-15707-f004:**
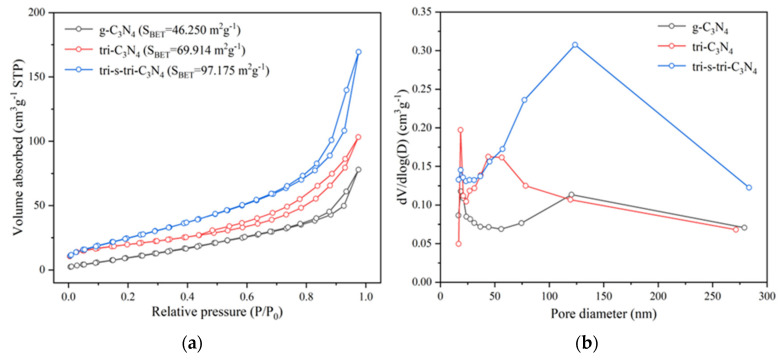
N_2_ adsorption–desorption isotherms (**a**) and corresponding pore-size distribution plots (**b**) of g-C_3_N_4_, tri-C_3_N_4_, and tri-s-tri-C_3_N_4_.

**Figure 5 ijerph-19-15707-f005:**
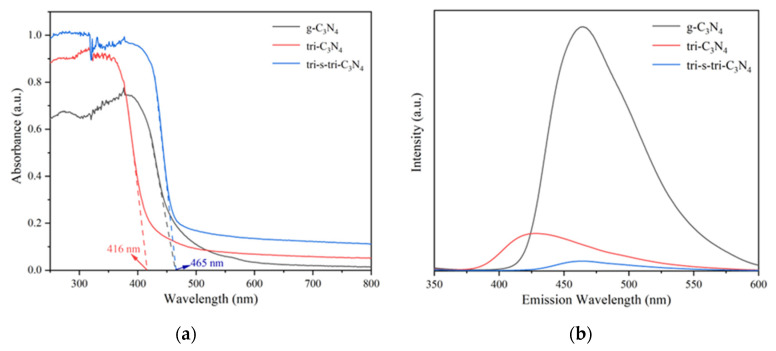
UV-vis/DRS spectra (**a**) and PL spectra (**b**) of g-C_3_N_4_, tri-C_3_N_4_, and tri-s-tri-C_3_N_4_.

**Figure 6 ijerph-19-15707-f006:**
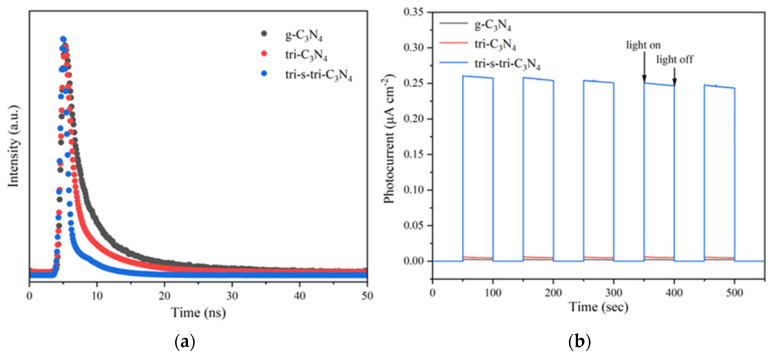
TR-PL spectra (**a**) and transient photocurrent response plots (**b**) of g-C_3_N_4_, tri-C_3_N_4_, and tri-s-tri-C_3_N_4_.

**Figure 7 ijerph-19-15707-f007:**
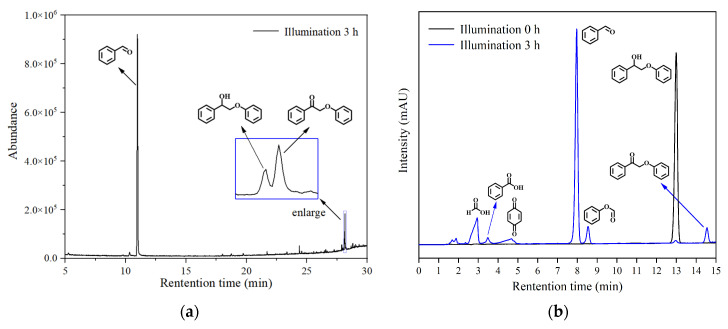
Overall GC-MS ion-flow diagram (**a**) and HPLC (**b**) of photocatalytic products of ML.

**Figure 8 ijerph-19-15707-f008:**
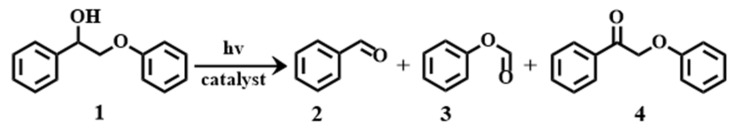
Schematic diagram of photocatalytic conversion of ML. (1–4 represented 2-phenoxy-1-phenylethanol, benzaldehyde, phenyl formate and 2-phenoxyacetophenone, representatively).

**Figure 9 ijerph-19-15707-f009:**
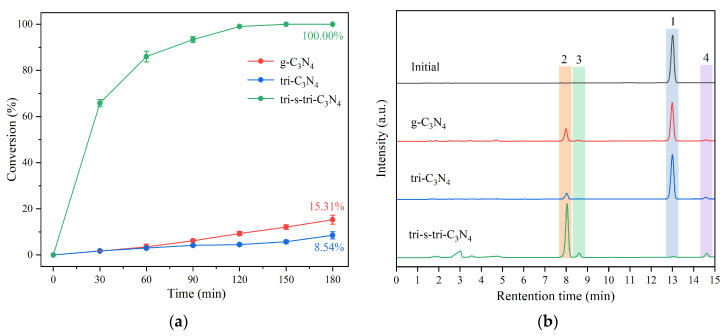
Conversion plot (**a**) and HPLC (**b**) of ML conversion using different carbon nitride photocatalysts at 3 h.

**Figure 10 ijerph-19-15707-f010:**
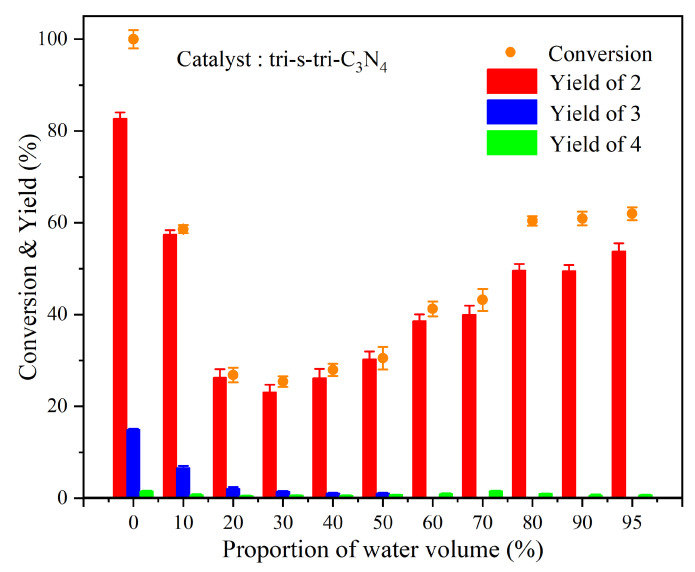
Photocatalytic conversion of ML using tri-s-tri-C_3_N_4_ in water–acetonitrile solvent and comparison chart of main product yield.

**Figure 11 ijerph-19-15707-f011:**
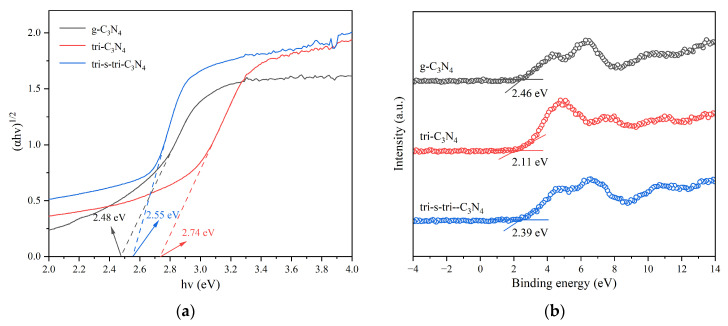
Kubelka–Munk diagram (**a**), XPS valence band map (**b**), Mott–Schottky plots (**c**), and bandgap structure (**d**) of materials.

**Figure 12 ijerph-19-15707-f012:**
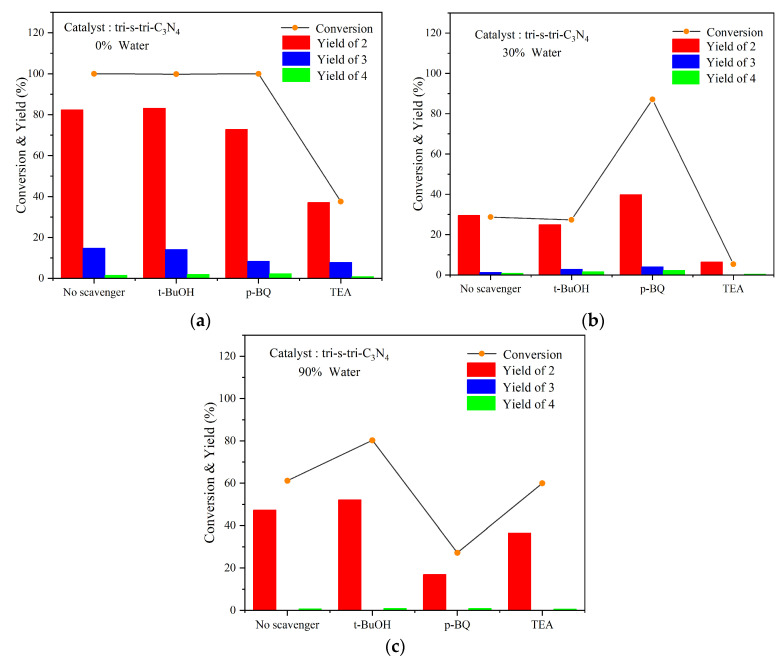
Effect of adding free-radical-quenching agents to (**a**) 0% water, (**b**) 30% water, and (**c**) 90% water with tri-s-tri-C_3_N_4_ as catalyst. (The concentration of the scavengers was 2 mM).

**Figure 13 ijerph-19-15707-f013:**
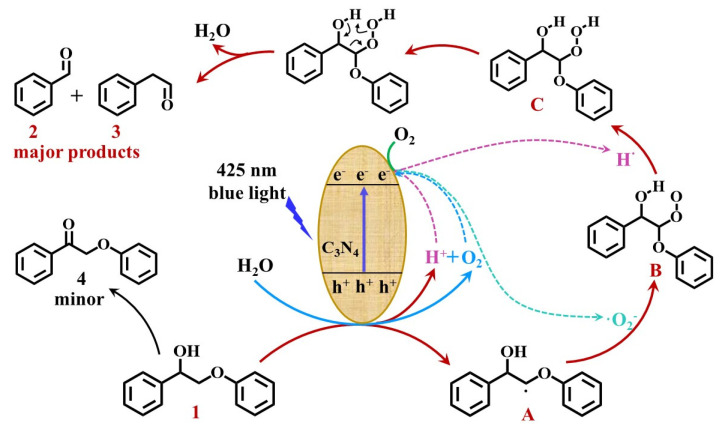
Conversion mechanism of ML in acetonitrile–water mixed solvents.

**Table 1 ijerph-19-15707-t001:** The results of the photocatalytic conversion of ML under different reaction conditions.

Group	Catalyst	Conversion of 1 (%)	Product Yield (%)			C_α_-C_β_ Cleavage Selectivity (%)
			2	3	4	
1	None ^a^	0	-	-	-	0
2	g-C_3_N_4_ ^a^	15.31	14.70	2.66	0.47	96.02
3	tri-C_3_N_4_ ^a^	8.54	8.47	1.18	0.71	99.20
4	tri-s-tri-C_3_N_4_ ^a^	100.00	84.90	14.05	1.52	84.90
5	tri-s-tri-C_3_N_4_ ^b^	8.37	8.00	2.42	0.35	95.58
6	tri-s-tri-C_3_N_4_ ^c^	40.70	36.12	6.58	1.95	88.75
7	tri-s-tri-C_3_N_4_ ^d^	36.80	35.33	6.57	2.29	96.01
8	tri-s-tri-C_3_N_4_ ^e^	1.24	-	-	-	0

Reaction conditions: 0.4 mmol/L **1**, 10 mg catalyst, 50 mL acetonitrile, 20 W LED (λ = 420–430 nm), room temperature, air atmosphere, reaction time 3 h; ^b^ reaction in methanol; ^c^ reaction in ethyl acetate; ^d^ reaction in acetone; ^e^ reaction conducted in the dark; “-” indicates not detected. The results are represented as the averaged data of three replicates.

**Table 2 ijerph-19-15707-t002:** Experimental results of photocatalytic conversion of ML in acetonitrile–water solvents with different volume proportions.

Group	Acetonitrile–Water Volume Ratio	Conversion of 1 (%)	Product Yield (%)			C_α_-C_β_ Cleavage Selectivity (%)
			2	3	4	
1	100–0	100.00	84.90	14.91	1.49	84.90
2	90–10	58.61	57.44	6.60	0.74	98.00
3	80–20	26.84	26.32	2.12	0.54	98.06
4	70–30	25.39	23.06	1.44	0.65	90.82
5	60–40	27.94	26.23	1.09	0.62	93.88
6	50–50	30.51	30.33	1.09	0.77	99.41
7	40–60	41.25	38.59	-	0.98	93.55
8	30–70	43.19	39.96	-	1.54	92.52
9	20–80	60.43	49.61	-	1.01	82.09
10	10–90	60.95	49.46	-	0.66	81.15
11	5–95	62.00	53.82	-	0.70	86.80

Reaction conditions: 0.4 mmol/L **1**, 10 mg catalyst, 50 mL solvent, 20 W LED (λ = 420–430 nm), room temperature, air atmosphere, reaction time 3 h; “-” indicates not detected; the results are represented as the averaged data of three replicates.

## Data Availability

Not applicable.

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
