# Peer review of "Selective Photocatalytic Transformation of Lignin to Aromatic Chemicals by Crystalline Carbon Nitride in Water–Acetonitrile Solutions"

_ijerph, 2022, doi:10.3390/ijerph192315707_

Round 1

Reviewer 1 Report

Abstract

-          The abstract seems to be more concision. It was focused on the photocatalytic conversion of lignin to aromatic compounds in the aqueous solution.

Introduction

-          It is noted that your manuscript needs careful enhancing by showing the novelty and the different in the work in compared to the previous similar works.

Results

-          It was recommended to make elemental analysis and mapping of the studied SEM images by EDX

-          It better to make discussion/ justification/ explanation in depth with assistant of XPS or Raman

Reviewer 2 Report

In this work, crystalline carbon nitride with a high visible light response was prepared by a simple two-step calcination process. The crystalline carbon nitride was applied to the selective conversion of lignin to aromatic compounds in an aqueous solution. Most of the conclusions are well data-supported. And it provides a theoretical basis for the green transformation of lignin in the aqueous solutions. I would like to recommend this work be accepted after the following points are addressed.

1. Please check the grammar and formatting in the manuscript and carefully check your word typing.

2. As you mentioned, " The mixtures were magnetically stirred in the dark for 30 minutes before turning on the light source, to reach adsorption equilibrium. " The mixtures still stirred during the irradiation? Because the C3N4 is not soluble in water or organic solvents (such as the acetonitrile you used) and may deposit down eventually without stirring.

3. The reaction conditions are not indicated under tables 1 and 2 in the manuscript.

4. The manuscript mentions Camellia oleifera shell lignin, why do you select it but not others? And this paper only studies lignin model compounds but not Camellia oleifera shell lignin as the substrate? Please specify.

5. In the elaboration of Figure 6a, “short-lived component”, “long-lived component” and other professional terms in the manuscript should be corrected.
